# Numerical Analysis and Optimization of the Front Window Visor for Vehicle Wind Buffeting Noise Reduction Based on Zonal SAS *k-ε* Method

**Zhendong Yang** [1,2,*]  **, Longgui Liu** [3] **and Zhengqi Gu** [2]

1   Department of Automotive Engineering, School of Mechanical and Electrical Engineering, Lingnan Normal University, Zhanjiang 524048, China
2   State Key Laboratory of Advanced Design and Manufacturing for Vehicle Body, Hunan University, Changsha 410082, China; guzhengqi63@126.com
3   School of Mechanical Engineering, Hunan University of Technology, Zhuzhou 412007, China; liulonggui@126.com
*   Correspondence: yangzhd@lingnan.edu.cn or yzdrly@hnu.edu.cn; Tel.: +86-759-3183976

**Abstract:** Numerical investigations were conducted to determine the effectiveness of the front window visor for wind buffeting noise reduction. An unsteady flow simulation was carried out using a zonal Scale Adaptive Simulation (SAS) *k-ε* turbulence model. Firstly, the accuracy of the simulation method was validated based on a benchmark problem. The benchmark results, frequency, and sound pressure levels of feedback and resonance modes all matched well with the experimental data. The effect of the front window on the buffeting noise reduction was numerically investigated based on three different front side window openings. The analysis focused on the suppression effect of the front window visor. The results show that the front window visor changed the A-pillar vortex shedding trajectory and thus reduced the driver's ear pressure fluctuation. On this basis, an optimization algorithm was employed to optimize the shape of the front window visor. The main design goal was to decrease the sound pressure level (*SPL*) values of the driver's left ear. Simulation results showed that the monitoring point's *SPL* of buffeting noise after the visor optimization was reduced by 12.6%, compared with that of the original visor.

**Keywords:** vehicle window buffeting noise; front side window visor; zonal scale adaptive simulation; optimization





## 1. Introduction

In recent years, a significant amount of research has been carried out on mechanical noise (such as engine noise, transmission noise, etc.), and effective noise control methods have been proposed; wind buffeting noise is becoming more and more prominent with the continuous increase in the practical speed of vehicles. Wind buffeting noise is caused by the coupling of flow instability of the shear layer and the air in the vehicle cabin while driving a vehicle with a sunroof or window open, and is a type of aerodynamic noise. Wind buffeting noise has the features of high strength and low frequency. It not only affects the vehicle ride comfort but also affects driving safety. Therefore, to improve the quality of automobile products, it is very important for automotive manufacturers to predict and analyze the wind buffeting noise at the designing stage.

The characteristics of sunroof buffeting noise and its control methods have been extensively studied [1–10]. However, there are few research findings on the characteristics and the control methods of buffeting noise caused by opening the side window. Yang et al. conducted road tests on the buffeting noise of a sedan caused by different side window openings [11]. The results showed that there exist discrepancies in the characteristics of wind buffeting noise. When a single rear window opens, or two rear windows open at

the same time, buffeting noise exhibits a feature of multi-harmonic oscillation, while in other cases only a single peak appears in the *SPL*'s spectrum. GU et al. summarized the simulation and control measures of vehicle window buffeting noise [12,13], such as installing a guide plate at the leading edge of the window, changing the opening size of windows, setting vents, etc., which promoted wind noise control technology in vehicle engineering. Utilizing CFD numerical simulation, An et al. proposed five control measures to reduce the wind buffeting noise of an SUV [14], the results showed that certain noise reduction can be achieved through five passive control methods. Balasubramanian et al. numerically studied the front wind buffeting noise characteristics caused by two mirror geometries: a glass-mounted mirror and a door-mounted mirror [15]. They found that the door mirror was noisier by approximately 6 dB. Door mirrors have a major impact on the wind noise observed at the driver's ear. Walker and Wei minimized buffeting by optimizing the mirror angle [16]. Any combination of open windows can suppress the buffeting of the side windows, and the validity of this method has been verified by numerical investigations on a car [17] and on an SUV [18]. He et al. installed a cylindrical deflector on the B-pillar to suppress the rear window buffeting noise [19]. By optimization using the response surface method, the reduction of peak *SPL* of buffeting noise was more than 12 dB in comparison with the original state without a deflector. These measures to suppress wind buffeting noise have achieved obvious results. However, the styling changes of some aerodynamic components are severely limited to the automotive design and the aerodynamic performance of the vehicle. Only minor shape changes are generally acceptable, and the choice of the car manufacturer is a suitable modification.

The rain visor of the front side window can effectively prevent rain or strong wind blowing directly into the vehicle cabin, thus improving the ventilation efficiency and preventing the window from fogging in the rain. In addition to these functions, the visor can also reduce the wind buffeting noise caused by opening the front window. However, the related literature has not been seen. In this paper, the effect of reducing wind buffeting noise is discussed through CFD numerical simulation, and then the numerical simulation optimization is carried out to find the best shape of the visor, which can offer a reference for theoretical research and engineering application of vehicle wind buffeting noise.

## 2. Computational Schemes

Scale adaptive simulation (SAS) is an improved URANS formulation based on introducing the von Karman scale into the turbulence scale equation [20]. The information provided by the von Karman length-scale allows SAS models to dynamically adjust to resolved structures in a URANS simulation, which results in a LES-like behavior in unsteady regions of the flow field. At the same time, the model provides standard RANS capabilities in stable flow regions. The computational resources consumed by scale adaptive simulation are only half that of the large eddy simulation [20], which is very suitable for engineering applications. However, under the condition of low Mach number flow, SAS cannot trigger the LES-like behavior in the flow region far from the wall due to the flow instabilities loss. In the flow region near the wall, it cannot show the scale resolution ability due to the defects of the RANS equation [21].

Given the advantages and disadvantages of the SAS turbulence model, combined with the widely used standard $k - \varepsilon$ turbulence model in engineering, according to the literature [22], the SAS equation can be rewritten as follows:

$$\frac{\partial(\rho k)}{\partial t} + \frac{\partial(\rho U_i k)}{\partial x_i} = G_k - \rho\varepsilon + \frac{\partial}{\partial y}\left[\frac{\mu_t}{\delta_k}\frac{\partial k}{\partial x_j}\right] + Y_M \tag{1}$$

$$\frac{\partial(\rho\varepsilon)}{\partial t} + \frac{\partial(\rho U_i \varepsilon)}{\partial x_i} = \frac{\partial}{\partial x_j}\left(\mu_t\frac{\partial\varepsilon}{\partial x_j}\right) + c_{1\varepsilon}\frac{\varepsilon}{k}G_k - c_{2\varepsilon}\rho\frac{\varepsilon^2}{k} + S_\varepsilon \tag{2}$$

where $k$ is the turbulence kinetic energy, $\varepsilon$ is the dissipation rate, $\mu_t$ is the turbulent viscosity, $G_k$ represents the generation of turbulence kinetic energy, $G_k = -\rho\overline{u_i u_j}\frac{\partial u_j}{\partial x_i}$, $Y_M = 2\rho\varepsilon M_T^2$ is the contribution of the fluctuating dilatation in compressible turbulence to the overall dissipation rate, $M_T = \sqrt{k/a^2}$ is the turbulent Mach number, and $a$ is the speed of sound. This compressibility modification is available when the compressible form of the ideal gas law is used. The model constants $c_{1\varepsilon}$, $\delta_k$, and $\sigma_\varepsilon$ have the following default values:

$c_{1\varepsilon} = 1.44$, $c_u = 0.09$, $\sigma_k = 1.0$, $\sigma_\varepsilon = 1.3$.

In Equation (2), the user-defined source term is:

$$S_\varepsilon = 0.47\frac{\sqrt{k}}{L_{v\kappa}}G_k - 2\frac{u_t}{\varepsilon}\left(\frac{\partial\varepsilon}{\partial x_j}\right)^2 + 5\frac{u_t}{k}\left(\frac{\partial k}{\partial x_j}\right)\left(\frac{\partial\varepsilon}{\partial x_j}\right) - \frac{\rho}{3}\left(\frac{\partial k}{\partial x_j}\right)^2 \tag{3}$$

The von Karman length-scale is redefined as:

$$L_{v\kappa} = \kappa\frac{\|S\|}{\|U''\|} \tag{4}$$

where $\kappa = 0.41$ is the von Karman constant. The first velocity derivative $\frac{\partial U}{\partial y}$ is represented in $L_{v\kappa}$ by $U'$, which is equal to $S_{ij}$, a scalar invariant of the strain rate tensor $S$: $U' = S = \sqrt{2 \cdot S_{ij}S_{ij}}$, $S_{ij} = \frac{1}{2}(\frac{\partial U_i}{\partial x_j} + \frac{\partial U_j}{\partial x_i})$.

The second velocity derivative, $U''$, is generalized to 3D using the magnitude of the velocity Laplacian:

$$U'' = \sqrt{\frac{\partial^2 U_i}{\partial x_k^2}\frac{\partial^2 U_i}{\partial x_j^2}} \tag{5}$$

The model also provides direct control of the high wave number damping. This is realized by a lower constraint on the value of $L_{vk}$ in the following way:

$$L_{vk} = \max(\kappa\frac{\|S\|}{\|U''\|}, C_s\sqrt{\frac{\kappa\eta_2}{(\beta/c_u) - a}} \cdot V^{1/3}) \tag{6}$$

To describe the two flow regions, a blending function for the $\varepsilon$ equation across the interface is used at the interface to describe both flow regions with the turbulence models of the same representation level. In the present study, the interface is placed at $Y^+ = 80$. According to the literature [23], two different values are adopted in the present approach: $c_{2\varepsilon} = 1.92 < 2$ is adopted in the near-wall region (low Reynolds number region), and $c_{2\varepsilon} = 2.36 > 2$ is adopted in the area away from the wall (high Reynolds number area).

## 3. Methodology and Validity of CFD Simulations

Yang et al. conducted road tests on the buffeting noise of a sedan caused by different side window openings [11]. The results showed that Helmholtz resonance plays an important role. Flow over an open side window in a vehicle exhibits similar characteristics to the flow over an open deep cavity. They all present the phenomenon of flow-excited Helmholtz resonance. A deep cavity as a benchmark problem for the validity of the CFD simulations is shown in Figure 1.

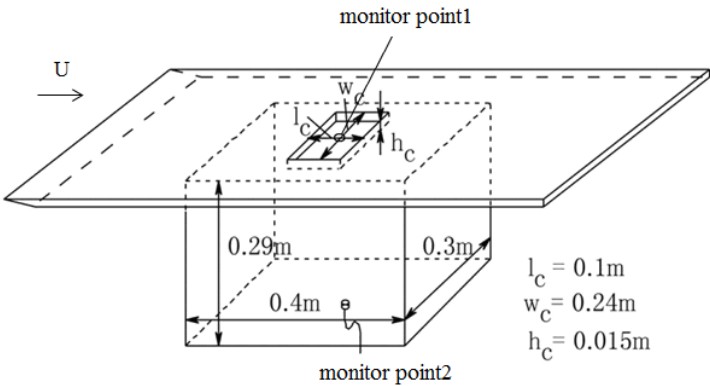

**Figure 1.** Cavity model and receiver location.

The dimensions are similar to those used in wind tunnel experiments. The methodology for obtaining these experimental results has been described in the literature [4,17]. Flow is fully turbulent over the cavity with an inlet-free stream velocity of 25 m/s. One receiver point, at the center of the cavity bottom, is considered to validate the frequency spectrum with experimental results. To ensure an accurate resolution of the shear layer dynamics, more meshes are allocated over the opening of the cavity. The minimum grid spacing around the cavity corners is set to 0.1 mm. The total number of node points is approximately $2.576352 \times 10^6$, and the number of hexahedral cells is $2.504167 \times 10^6$. The total number of elements is $2.647365 \times 10^6$. Partial grids along the longitudinal symmetry plane and prism layer are shown in Figure 2.

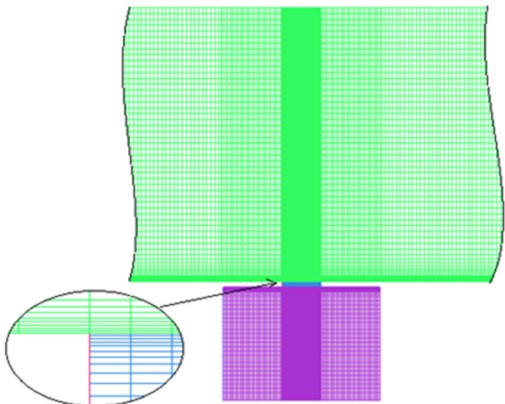

**Figure 2.** Partial grids along the longitudinal symmetry plane and prism layer.

To assess the reliability of the numerical solution, a grid dependence study was carried out, as shown in Table 1. The mean pressure coefficient and the peak values of the second monitor point's pressure fluctuation have been extracted from the simulation results for the three different mesh refinements. From Table 1, the discrepancies for the receiver location's mean pressure coefficient are very small. Furthermore, the difference between the peak pressure fluctuation for the fine and medium mesh is not large. Therefore, a fine mesh was used in this study.

**Table 1.** Mean pressure coefficient and peak value of the pressure fluctuation for monitor point 2.

| Grid | Number of Elements | Mesh Size (mm) | | | | | | Mean Pressure Coefficient | Peak Value of the Pressure Fluctuation |
|------|-------------------|----------------|----------------|----------------|----------------|----------------|----------------|---------------------------|----------------------------------------|
|      |                    | $x_{min}$ | $x_{man}$ | $y_{min}$ | $y_{man}$ | $z_{min}$ | $z_{max}$ |                           |                                        |
| Coarse | $1.844382 \times 10^6$ | 2.60 | 19.12 | 4.06 | 10.21 | 1.00 | 20.60 | 0.05159 | 92.75 pa |
| Medium | $2.247365 \times 10^6$ | 2.30 | 15.85 | 3.03 | 6.21 | 0.50 | 18.81 | 0.05165 | 87.31 pa |
| Fine | $2.647365 \times 10^6$ | 2.00 | 10.21 | 2.42 | 4.82 | 0.10 | 15.27 | 0.05171 | 86.56 pa |

Five monitoring lines were selected in the flow direction (x) of the opening area of the cavity; the height of each line is 0.9 m, and the distances from the front edge of the opening of the cavity are 0.01 m, 0.03 m, 0.05 m, 0.07 m, and 0.09 m. The streamwise direction length of the opening of the cavity ($l_c$ = 0.1 m) is used for dimensionless treatment. In the simulation, the mean streamwise velocity of the monitoring point is obtained through calculation. The comparison between the wind tunnel test results and the simulation results is shown in Figure 3. The abscissa adopts the ratio of the mean velocity ($U_{\text{mean-x}}$) to inlet velocity ($U_0$ = 25 m/s) for dimensionless treatment, and the ordinate takes the ratio of height (z) in the vertical direction to the opening height of the cavity ($10h_c$ = 0.15 m) for dimensionless treatment. It can be seen from Figure 3 that the trend of simulated values is consistent with the experimental values at different locations, and the shape of the mean velocity curve approximately matches with the inverse hyperbolic tangent function distribution, which is consistent with the literature [24]. This proves that the zonal SAS $k$-$\varepsilon$ turbulence model can accurately capture the flow characteristics.

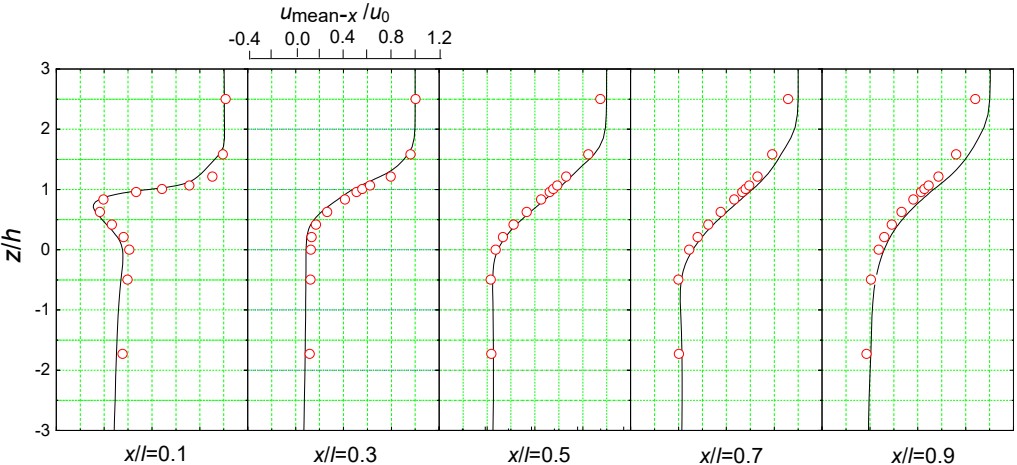

**Figure 3.** Mean streamwise velocity profile at the cavity opening. Note: circles represent experimental values and solid lines represent simulated values.

Based on the time histories of the fluctuating pressure, it is possible to convert the computed data to the frequency-domain results by a standard fast Fourier transform (FFT) routine. The sound pressure level (*SPL*) was finally converted to dB units using the formula:

$$SPL = 20 \log \frac{p}{p_{ref}} \tag{7}$$

where $p$ is the fluctuating pressure and $p_{ref}$ is the sound pressure for reference:

$$p_{ref} = 2 \times 10^{-5} \text{pa} \tag{8}$$

Figure 4 displays the computed and measured sound pressure level at monitor point 2 for the inlet flow speed of 25 m/s.

Four resonant frequencies—101.0 Hz, 202.2 Hz, 303.3 Hz, and 403.3 Hz—could be easily identified from the experimental data. For the simulation results, four resonant peaks—102.8 Hz, 204 Hz, 308 Hz, and 408 Hz—are also shown in Figure 3. The first resonant peak is the fundamental frequency of the flow instability over the cavity. For the frequency of the first peak *SPL*, there is no significant difference between the experimental results and the simulation results; the difference is less than 2 Hz. The frequency of the second peak *SPL* captured differs from the experimental value by 2 Hz, within acceptable ranges. Both the frequencies of the third peak *SPL* and the fourth peak *SPL* are further and further away from the test values. The overall trend of the *SPL* spectra captured by the zonal SAS $k - \varepsilon$ turbulence model almost perfectly matches with the experimental

values. The present model underestimates the amplitude of the high-order resonance and overestimates the frequency of the high-order resonance, which may be explained by the following. First, these were caused by the difference between the numerical simulations and the experimental measurements. It is very important to note in our numerical simulations that a uniform velocity profile with a turbulent intensity of 0.5% was assumed at the inlet boundary of the flow field. However, the turbulent boundary expected in the experimental measurements was generally not available. Second, the computation model used in the present study did not include the simulation of the background noise. Third, this might be due to the fact that the higher-order modes cause relatively weak oscillation in the present simulation.

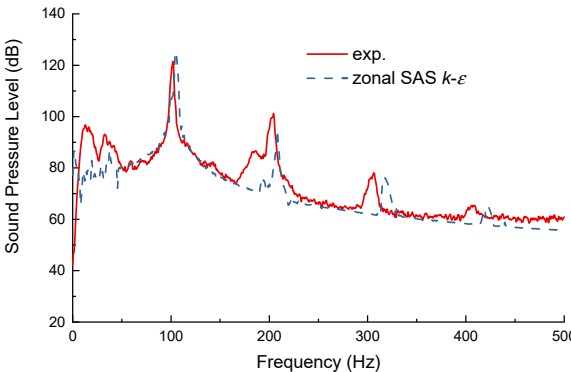

**Figure 4.** Comparison between calculation results and test results at monitor point 2.

In general, the agreement between the compared results is very satisfactory, showing the ability of the zonal SAS $k - \varepsilon$ model to predict the cavity noise. It indicates that this model can be used for window buffeting noise predictions at earlier stages of the program and verification of the design of the buffeting fix.

## 4. Effects of the Rain Visor on the Reduction of the Window Buffeting Noise

The numerical simulation model of the sedan refers to the literature [17]. The meshing scheme is adopted from the best practices of Scale Resolved Simulation (SRS) [25]. The numerical simulation model of the sedan is appropriately simplified for reducing the calculation, and the external flow field of the screen wiper, door handles, and other accessories are ignored. The unstructured computational grids used in the simulations were constructed with ICEM CFD 16.2. The total size of the mesh for the computational domain is $8.484437 \times 10^6$. Installing the rain visor on the A-pillar, the surface meshes of the left front window are shown in Figure 5.

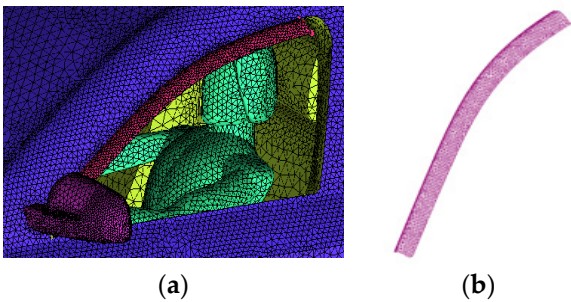

(**a**)  (**b**)

**Figure 5.** The surface mesh of the left window with the rain visor. (**a**) The window; (**b**) The visor.

As shown in Figure 6, when the left front window is fully open, there are two vorticity lines in the shear layer of the horizontal tangential plane through the driver's ear. The outer side vortex is shedding from the upper edge of the rearview mirror, and the inner side vortex is from the A-pillar. When the A-pillar is installed with the visor, the trajectory

of the vortex from the A-pillar is changed, and the amount of vorticity invading the cabin is reduced. Therefore, the fluctuation pressure beside the driver's left ear is reduced, as shown in Figure 7a.

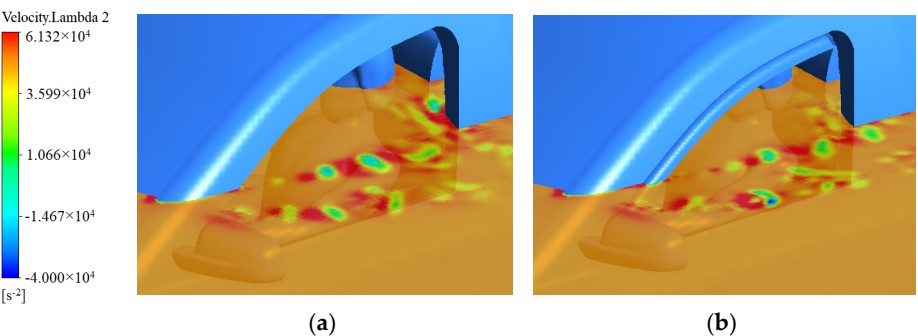

**Figure 6.** Vorticity on the horizontal plane through the driver's ear. (**a**) Without the visor; (**b**) With the visor.

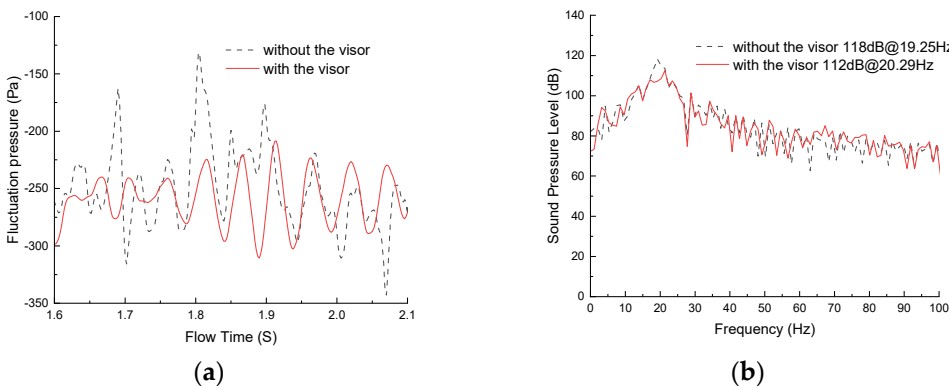

**Figure 7.** Fluctuation pressure and sound pressure level at the driver's left ear with the fully opened left front window. (**a**) Fluctuation pressure; (**b**) Sound pressure level.

In the case of no visor, the peak *SPL* of the driver's left ear is 118 dB; the peak *SPL* is reduced by 6 dB after installation of the visor, and the corresponding frequency is shifted from 19.25 Hz to 20.29 Hz, as shown in Figure 7b. Therefore, the rain visor can effectively reduce the wind buffeting noise when the front window is fully opened.

Therefore, the wind buffeting noise caused by the front window opening can be effectively suppressed by adding a rain visor to the A-pillar.

## 5. Optimization of the Rain Visor on the Reduction of the Window Buffeting Noise

### 5.1. Design Variables and Constraints

In this research, the front rain visor of a sedan is selected as the optimization object, and the optimization parameters are shown in Figure 8.

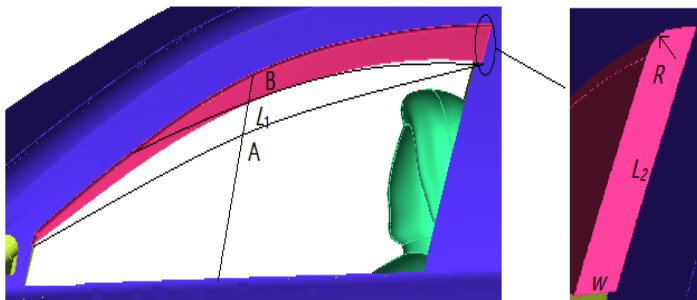

**Figure 8.** Optimization parameters of the visor.

According to the modelling features of the rain visor, the following parameters were selected as design variables without affecting the driver's field of vision:

(1) The design variable $L_1$: Take the midpoint of the upper and lower sides of the front window and make the midline on the window's curved surface. Make the intersection midline of the two endpoints of the rain visor on the window's curve at one point, then the curve will be regarded as the edge line of the rain visor, and the intersection point is the radiant point of the rain visor's edge. Considering the rain shielding effect of the rain visor and the range of visual field of the driver, two points, A and B, are taken as the limit position of the intersection, and the curve between them is $L_1$. Make the position of point B zero, and then the value range of $L_1$ is [0~39 mm].

(2) The design variable $W$: $W$ is the distance between the lower edge of the visor and the window's curved surface. The slope of the noise reduction visor's surface can be controlled by changing the value of W. When the slope of the visor's surface is greater than 45°, the rain shielding effect will decrease sharply; set the minimum value of $W$ as 20 mm, so the value range of $W$ is [20~36 mm].

(3) The design variable $L_2$: Set the length of the long side of the visor, i.e., the length of the rear side of the B pillar, as $L_2$. According to the requirements of the radian of the window, the value range of $L_2$ is set as [50~65 mm].

(4) The design variable $R$: The radius of the rounded angle of the visor is set as $R$. Restricted by the short edge of the visor, the value range of $R$ is set as [0~15 mm].

According to the previous analysis, the front visor can change the trajectory of the vortex in the shear layer separated from the A-pillar. Therefore, when the left front window is fully open and the inlet velocity is 30 m/s, the peak value of the sound pressure level at the driver's left ear is taken as the objective function, $P$, and the objective function value cannot exceed 116 dB.

### 5.2. Approximate Agent Model

According to the value range of design variables, four design variables were selected, including the upper and lower edges' midline $L_1$ of the window, the distance, $W$, between the lower edge of the rain visor and the window, the long edge, $L_2$, of the visor, and the radius, $R$, of the visor. Ten groups of sample points are designed using the optimal Latin hypercube sampling method. The design scheme is shown in Table 2.

**Table 2.** Optimal Latin hypercube design of experiments.

|  | $R$ (mm) | $W$ (mm) | $L_1$ (mm) | $L_2$ (mm) |
|---|---|---|---|---|
| YU1 | 11.6 | 16.88 | 36.95 | 63.29 |
| YU2 | 5.00 | 30.67 | 16.42 | 53.95 |
| YU3 | 0 | 24.7 | 34.89 | 63.42 |
| YU4 | 9.83 | 22.66 | 2.05 | 58.68 |
| YU5 | 12.63 | 16.88 | 24.63 | 52.37 |
| YU6 | 12.42 | 26.69 | 32.84 | 50.79 |
| YU7 | 15 | 28.66 | 10.26 | 60.26 |
| YU8 | 9.97 | 14.66 | 12.64 | 57.14 |
| YU9 | 3.32 | 14.67 | 4.11 | 61.25 |
| YU10 | 1.58 | 20.66 | 6.16 | 53.16 |

According to ten groups of sample points in Table 2, CAD modelling was performed, the mesh was built, and ten groups of response values were obtained by CFD simulation calculation. The sound pressure level spectrums of the driver's left ear are shown in Figure 9. The sound pressure levels at the monitoring points of the driver's left ear have different degrees of decline. The fifth group of sample points is the best, with the lowest peak sound pressure level, as shown in Figure 9e.

Figure 10 shows the main effect of each design variable on the objective function value, $P$. It can be seen from the slope of different curves in Figure 10 that the peak value of the

wind buffeting noise inside the vehicle cabin decreases with the increase in $L_1$ and $W$. In comparison, the slope of $L_1$ is greater than $W$, so $P$ changes more significantly with $L_1$. Meanwhile, as $R$ and $L_2$ increase, $P$ increases and then decreases.

The kriging model can reduce the computational cost of performing optimization [26]. According to the optimal Latin hypercube design of experiments in Table 2 and the CFD results in Figure 9, the kriging model was used to establish an approximation model of the response relationship between the design variables and the optimization objective. To verify the accuracy of the approximation model, any two test points outside the scheme of the design of the experiment are selected in the design space for CFD simulation, compared with the results obtained by the approximation model. The comparison results are shown in Table 3. It can be seen from Table 3 that the results calculated by using the approximation model are quite close to those obtained by CFD simulation. The results show that the established approximation model can well describe the relationship between design variables and response. The established approximation model has high precision and can directly replace CFD simulation.

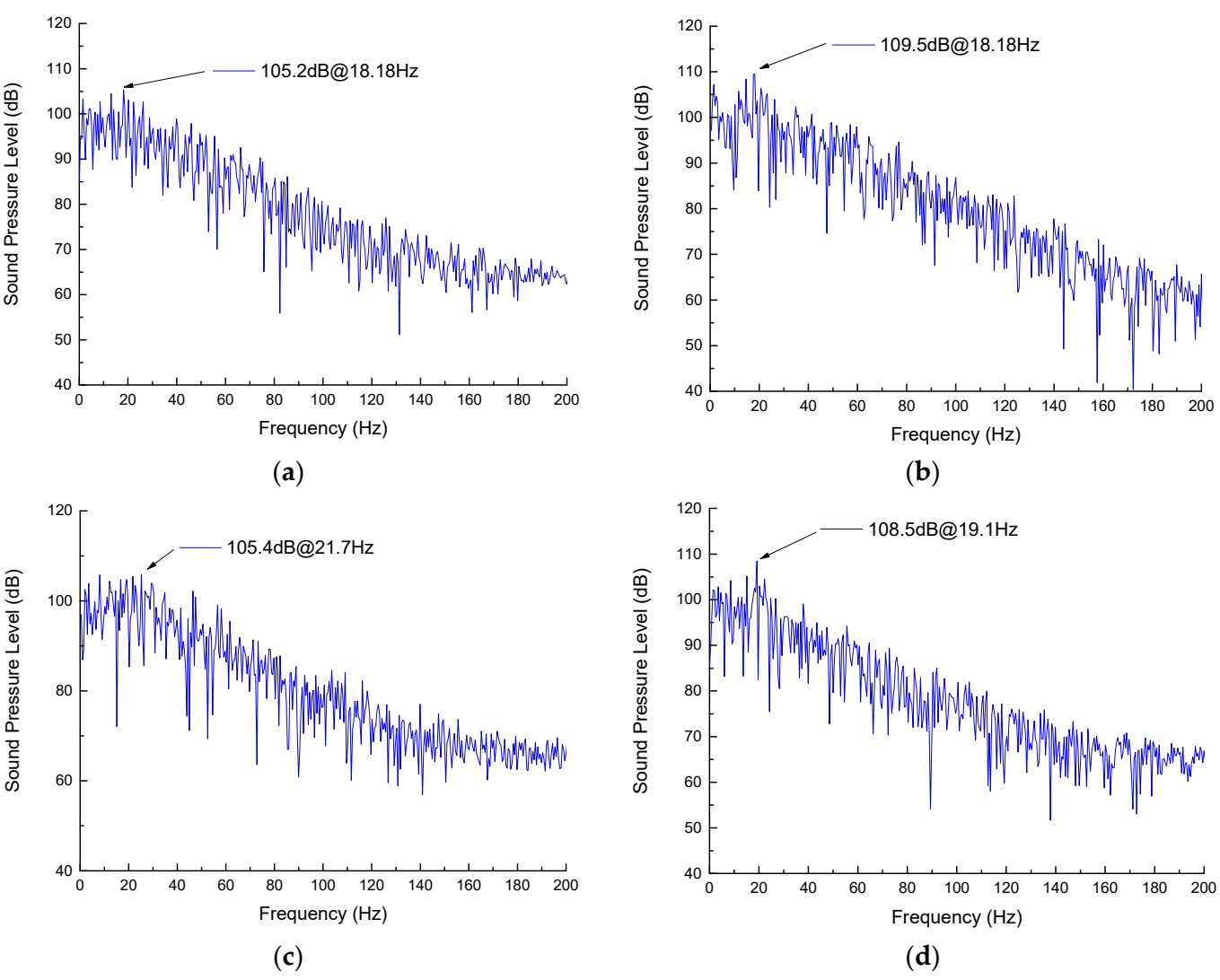

**Figure 9.** *Cont.*

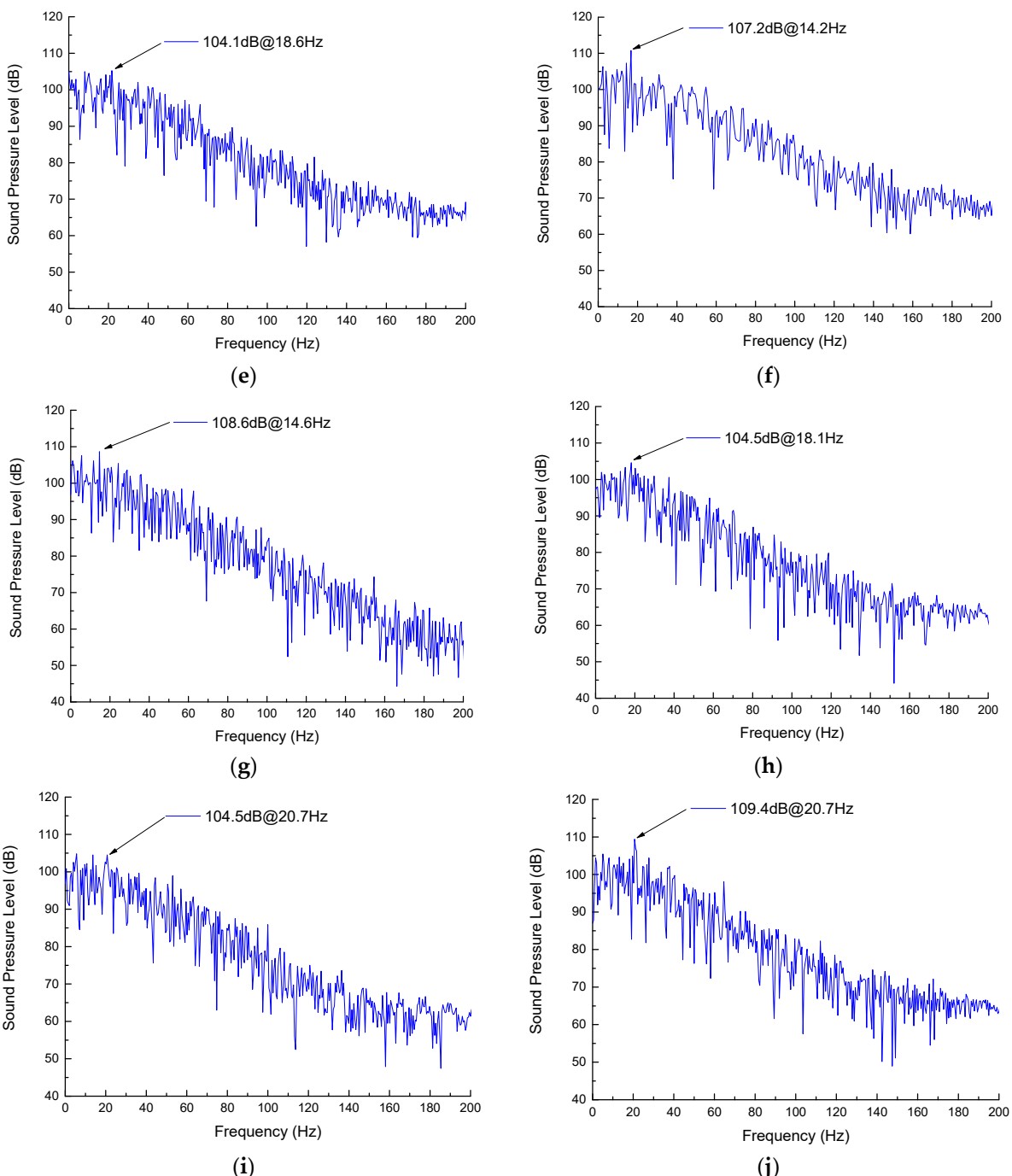

**Figure 9.** The sound pressure level spectrum at the driver's left ear. (**a**) YU1; (**b**) YU2; (**c**) YU3; (**d**) YU4; (**e**) YU5; (**f**) YU6; (**g**) YU7; (**h**) YU8; (**i**) YU9; (**j**) YU10.

**Table 3.** Analysis of fitting accuracy of approximation model.

| Number | Design Variable (mm) | | | | Peak Value *SPL* (dB) | | |
|---|---|---|---|---|---|---|---|
| | *R* | *W* | *L₁* | *L₂* | Approximation Model | CFD Simulation | Relative Error |
| 1 | 8 | 15 | 22 | 60 | 109.24 | 110.53 | 1.17% |
| 2 | 9 | 16 | 10 | 55 | 113.45 | 115.21 | 1.53% |

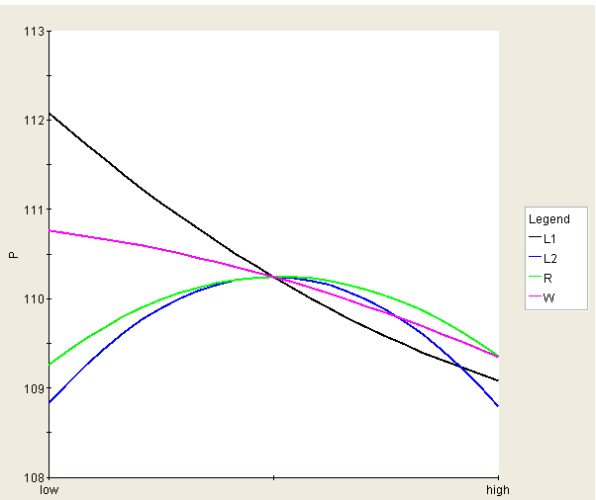

**Figure 10.** The main effect of each design variable on the objective function.

### 5.3. Global Optimization

Based on the approximation model, the multi-island genetic algorithm is applied to the global optimization design. The subgroup size is set to 20, the total population size is 50, and the total evolutionary algebra is 20 generations. The optimal solution is validated by CFD simulation. The comparison results are shown in Table 4. The noise reduction effect of the optimized visor is shown in Figure 11. It can be seen from Figure 11 that the vorticity invading the cabin is very small, and the peak sound pressure level at the driver's left ear is 102 dB. Compared with that without the visor of the front window, the noise value at the driver's left ear is reduced by approximately 14 dB, and the noise reduction range is 12.4%.

**Table 4.** Comparison of optimization results.

| Without the Visor | Approximation Model | Optimized Visor CFD Simulation | Error | Improvement Effect |
|---|---|---|---|---|
| 116.5 dB | 98.2 dB | 102 dB | 3.87% | −12.4% |

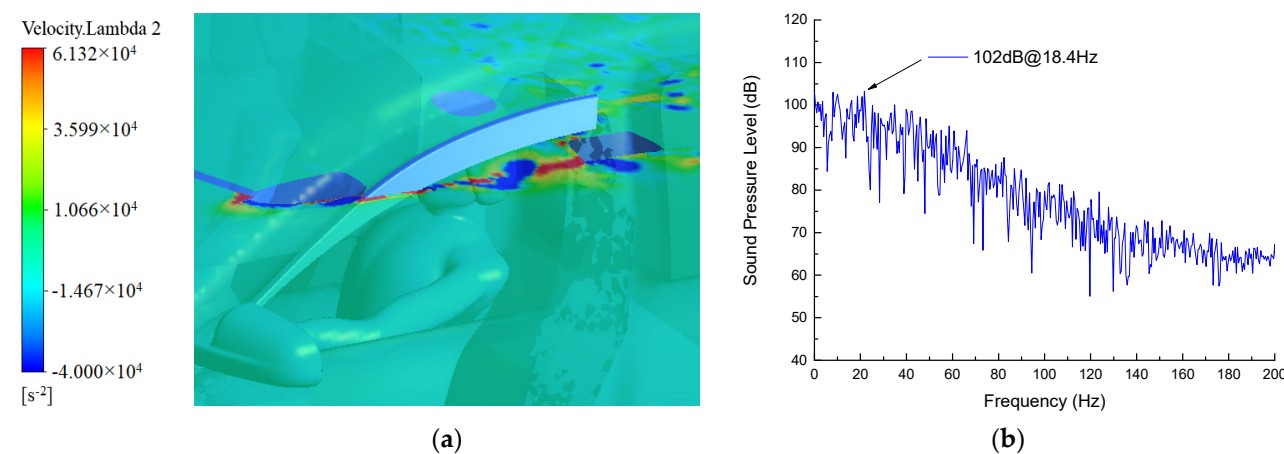

**Figure 11.** Noise reduction effect of the optimized visor. (**a**) The contour of velocity lambda2 at the horizontal section of the driver's ear; (**b**) Sound pressure level spectrum at the driver's left ear.

## 6. Conclusions

The phenomenon of the window buffeting noise caused by opening the front left window was analyzed by CFD. The results obtained are summarized as follows:

(1) A zonal formulation of the SAS $k - \varepsilon$ approach was employed in this study. A deep cavity as a benchmark problem was used to validate the approach. The results show that the CFD method can be used for side window buffeting noise predictions at earlier stages of the program.

(2) Aiming at the problem of the wind buffeting noise of the front side window, the CFD numerical simulation of the front side window fully opened was carried out. The results show that the front window visor changes the path of vortex shedding from the A-pillar, and thus reduces the vortex invading the cabin. The fluctuation pressure at the driver's ear is reduced, so the wind buffeting noise is better suppressed.

(3) The optimization algorithm is used to optimize the shape of the front rain visor. Compared with the case without the visor, the CFD results show that the optimized visor reduces the peak sound pressure level of the driver's left ear by 14.5 dB.

In future, validation of the reduction noise effect of the visor is needed via wind tunnel experiments and road tests.

**Author Contributions:** Conceptualization, Z.Y.; methodology, Z.Y.; validation, Z.Y.; investigation, Z.Y.; front window visor prototype development, Z.Y. and L.L.; supervision, Z.G. All authors have read and agreed to the published version of the manuscript.

**Funding:** This research was funded by Lingnan Normal University Scientific Research Project Funding (grant number ZL2027), State Key Laboratory of Automotive Simulation and Control of China (grant number 20181111), and the National Natural Science Foundation of China (grant number 51875186). The corresponding author would like to thank the China Scholarship Council for its financial support (CSC [2020] No. 50).

**Institutional Review Board Statement:** Not applicable.

**Informed Consent Statement:** Not applicable.

**Data Availability Statement:** All the data produced in this study is contained in the manuscript text.

**Acknowledgments:** The authors would also like to thank the reviewers for their valuable suggestions and kind work.

**Conflicts of Interest:** The authors declare no conflict of interest.

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
