# Peer review of "Numerical Analysis and Optimization of the Front Window Visor for Vehicle Wind Buffeting Noise Reduction Based on Zonal SAS k-ε Method"

_applsci, doi:10.3390/app12146906_

Round 1
Reviewer 1 Report
The manuscript describes a numerical work about vehicle wind buffeting noise reduction. Optimisation of the wind visor was performed based on zonal SAS k-epsilon method.
The work is interesting, but I could not recommend the publication due to a shortcoming in the method. Although the work provided a comparison with experiment in its validation before further modelling of the wind visor, the validation does not include a convergence study of the mesh adopted. It is utmost important for a numerical model to study the convergence of the results using different mesh resolution. This helps establish the credibility of the results.
Author Response
Response: Thank you for your suggestion. We would like to thank the reviewer for his/her interest and time in assessing our manuscript. It is of great appreciation to receive the thoughtful comments and suggestion made by the reviewer. Based on them, we revised our manuscript in order to improve its content and quality. If there are any further change needed, please let us know. we will be happy to address them.
We encompass the response to the raised issues, please see attached.

Reviewer 2 Report
Comments are introduced on the paper file that is attached.

Author Response
Thank you for your suggestion. We would like to thank the reviewer for his/her interest and time in assessing our manuscript. It is of great appreciation to receive the thoughtful comments and suggestion made by the reviewer. Based on them, we revised our manuscript in order to improve its content and quality. If there are any further change needed, please let us know. we will be happy to address them.
We submit our revised manuscript, please see attached.

Reviewer 3 Report
The manuscript addresses the numerical analysis and optimization of the from window visor of a vehicle by means of the zonal SAS k-ε method.
The introduction gives a clear idea of the current state of the art on this topic, with relevant references. These references are not completely up to date, as the newest one is from 2016.
The methods are adequately described, and the results are well presented, providing a good support for the statements of the text and the conclusions of the manuscript.
Nevertheless, some aspects must be addressed prior to the acceptance and publication of this manuscript:
1. Line 113: “Flow over an open side window in a vehicle exhibits similar characteristics as the flow over an open deep cavity”. Could the authors indicate the similarities in the manuscript?
2. Figure 3: What does the black dot represented in the subplot with x/l=0.9 mean?
3. Figure 4 shows experimental results compared to the zonal SAS k-ε method. Could the authors explain the methodology for obtaining these experimental results have been obtained?
4. Line 154: The first resonance associated to the fundamental frequency of the flow instability over the cavity is accurately detected by the numerical model. Nevertheless, as the authors also state, the difference between the numerical and experimental results obtained differs with the order of the resonance. Could the authors explain the cause? Is it due to the model itself or numerical errors in the simulations?
5. Section 4: What are the quality criteria and characteristics adopted for the mesh? Size, number of elements per wavelength, type of elements (quad, tetra, etc.).
6. Figure 5: It can be understood from Figure 6 and basics of automotive design what is the rain visor. Nevertheless, it might be interesting for some readers with less background in this field that this element is clearly marked in the figure.
7. References: As it was previously mentioned, the references are not completely up to date, as the newest one is 8 years old. I suggest including some more relevant and update references on the topic in order to know the most relevant works that have been recently published on this field.
Author Response
Thank you for your suggestion. We would like to thank the reviewer for his/her interest and time in assessing our manuscript. It is of great appreciation to receive the thoughtful comments and suggestion made by the reviewer. Based on them, we revised our manuscript in order to improve its content and quality. If there are any further change needed, please let us know. we will be happy to address them.
Please see the attachment.
Please see the attachment.

Round 2
Reviewer 1 Report
I would like to thank authors for addressing my concerns. The final form is good, but the mesh parameters of different mesh such as minimum mesh sizes and maximum mesh sizes in all directions should be included in Table 1.
Author Response
Thank you for your suggestion. We would like to thank the reviewer for his/her interest and time in assessing our manuscript. It is of great appreciation to receive the thoughtful comments and suggestion made by the reviewer. Based on them, we revised our manuscript in order to improve its content and quality. If there are any further change needed, please let us know. We will be happy to address them.
Please see the attachment.
